# Depletion of Caspase-12 Alleviates Retinal Degeneration in Aged BALB/c Mice Following Systemic Neonatal Infection by Murine Cytomegalovirus (MCMV)

**DOI:** 10.3390/v17091206

**Published:** 2025-09-03

**Authors:** Jinxian Xu, Xinyan Zhang, Yi Liao, Ting Shi, Brendan Marshall, Ming Zhang

**Affiliations:** 1Department of Cellular Biology and Anatomy, Medical College of Georgia, Augusta University, Augusta, GA 30912, USA; jinxu@augusta.edu (J.X.); xinyan.zhang@yale.edu (X.Z.); yiliao@augusta.edu (Y.L.); tshi@augusta.edu (T.S.); bmarshall@augusta.edu (B.M.); 2James and Jean Vision Discovery Institute, Medical College of Georgia, Augusta University, Augusta, GA 30912, USA

**Keywords:** caspase-12, murine cytomegalovirus, latency, p53, NF-κB, apoptosis, necroptosis

## Abstract

(1) Background: Retinal degeneration develops upon caspase-12 activation in aged BALB/c mice following systemic neonatal infection. (2) Methods: MCMV or medium was injected intraperitoneally (i.p.) into caspase-12^−/−^ and caspase-12^+/+^ mice (on BALB/c background) at <3 days after birth. At 8 and 12 months post infection (p.i.), eyes were analyzed by SD-OCT before eyes and extraocular tissues were collected and analyzed by plaque assay, H&E staining, TUNEL assay, Western blot and real-time RT-PCR. (3) Results: Virus DNA, but not replicating virus, was present in eyes and extraocular tissues at 8 and 12 months p.i. Several MCMV genes were expressed in eyes of both MCMV-infected caspase-12^−/−^ and caspase-12^+/+^ mice, while mean retinal thickness was significantly higher in MCMV latently infected aged caspase-12^−/−^ mice compared to age-matched infected caspase-12^+/+^ mice. Although similar levels of cleaved caspase-1 were detected in eyes of both infected caspase-12^−/−^ and control mice, significantly higher levels of activated NF-κB, cleaved caspase-8, MLKL, p-RIP3 and p53 were observed in eyes of infected caspase-12^+/+^ mice compared to eyes of infected caspase-12^−/−^ mice. (4) Conclusions: Our results suggest that caspase-12 contributes to retinal degeneration during MCMV ocular latency via multiple pathways including apoptosis and necroptosis.

## 1. Introduction

Human cytomegalovirus (HCMV), a β-Herpes virus subfamily with enveloped double-stranded DNA, infects 50 to 80% of humans [1]. Although generally asymptomatic, or a mild, transient illness in immunocompetent hosts, HCMV infection presents a serious threat to immunosuppressed (IS) or immunonaive individuals [2]. Following primary infection, HCMV persists for the life of the host through cycles of latency/reactivation [3], with multiple host sites/cell types, including endothelia and hematopoietic cells [4], susceptible to latency. The choroid/retinal pigment epithelium (RPE) might also be a site of HCMV latency in the eye, since studies of ocular tissue from human cadavers carried out in our laboratory revealed that HCMV DNA was present in 4 of 24 choroid/RPE samples [5]. HCMV infection is also associated with chronic vascular pathologies [6,7,8,9,10,11,12] and may be a risk factor for AMD progression, since there is a significant association between higher anti-HCMV IgG titers and neovascular AMD compared to dry AMD and non-AMD controls [13]. Since CMVs are strictly species-specific and HCMV cannot be studied experimentally in vivo, murine cytomegalovirus (MCMV) infection has been widely studied, both by ourselves and others, as a model system to decipher the mechanism of HCMV-induced ocular pathology [14,15,16,17,18,19,20,21,22,23,24,25,26,27].

Caspase-12 is a major inflammatory caspase, participating in innate immune responses and inflammation [28,29,30,31,32,33] as well as ER stress-related cell death [34,35,36]. Its role in innate immune responses against pathogens is controversial, due to the variety of pathogens and tissue/cell types which have been used for its study [28,29,30,31,32,33]. Caspase-12 resides on the outer ER membrane, where it can be activated following ER stresses, and although the mechanism remains unclear, activated caspase-12 is generally recognized as an inducer of ER stress-associated cell death [37].

The eye is one of several tissues/organs in which caspase-12 protein is constitutively expressed [37] and our previous studies have shown that caspase-12 is activated during MCMV retinitis [38,39,40]. Not only does Caspase-12 play a significant role in bystander retinal cell death but it also participates in the innate immune response against acute retinal virus infection [40]. Recent studies in our laboratory have shown that during systemic neonatal MCMV infection of BALB/c mice, MCMV disseminates to and remains latent in the choroid and RPE. Latent ocular infection combined with ocular MCMV gene expression induces in situ inflammation and development of retinal degeneration in aged mice [41]. Therefore, the purpose of this study was to determine if caspase-12 plays a role in retinal degeneration of aged latently infected BALB/c mice following systemic neonatal infection.

## 2. Materials and Methods

### 2.1. Virus

MCMV strain K181 was originally supplied by Dr. Edward Mocarski (Emory University, Atlanta, GA, USA). Viral titers were determined by plaque assay and virus stocks were prepared from salivary glands of IS BALB/c mice infected with MCMV as described previously [42].

### 2.2. Reagents and Antibodies

DNA primers were ordered from IDT (Coralville, IA, USA). RT^2^ Profiler™ PCR Array Mouse Antiviral Response (Cat#330231, GeneGlobe ID: PAMM-122Z), RT2 First Strand Kit (Cat# 330401) and RT2 SYBR^®^ Green FAST Mastermixes (Cat# 330620) were purchased from Qiagen (Germantown, MD, USA). Anti-RiP3 (#95702), MLKL (#37705), NF-κB p65 (#8242), NF-κB p52 (#4882), cleaved caspase-3 (#9664), cleaved caspase-8 (#8592), beta-actin (#3700), caspase-12 (#2202), LC3b (#3868), AKT (#9272), p-AKT (#4060), XIAP (#2042), p53 (#2524) and CHOP (#2895) were purchased from Cell Signaling Technology (Danvers, MA, USA). Anti-p-RIP3 (ab195117) was purchased from Abcam (Waltham, MA, USA). Caspase-1 (#06-503) was purchased from Millipore sigma (St. Louis, MO, USA). Goat anti-rabbit IgG HRP (sc2004) and rabbit anti-goat IgG HRP (sc2768) were from Santa Cruz (Santa Cruz Biotechnology, CA, USA). Anti-Mouse IgG HRP Conjugate (W402B) was purchased from Promega (Madison, WI, USA).

### 2.3. Mice

C57BL/6 caspase-12^−/−^ mice were purchased from Mutant Mouse Regional Resource centers (MMRRC) (Stock No: B6.129P2-Casp12^tm1Dgen^/Mmnc) and the derivation of these mice has been previously reported [28]. The experimental mice were obtained by crossing C57BL/6 caspase-12^−/−^ mice with BALB/c mice from Jackson Laboratory (Bar Harbor, ME, USA) for at least 7 generations. All mice were given unrestricted access to food and water and were maintained on a 12 h light cycle alternating with a 12 h dark cycle. The infected mice were housed in the BCL2 facility. All experiments and breeding followed the guidelines of the National Institutes of Health for the care and maintenance of mice and adhered to the ARVO Statement for the Use of Animals in Ophthalmic and Vision Research. The protocols were also approved by the Institutional Animal Care and Use Committee of Augusta University. All mice tested negative by PCR for the rd8 mutation.

### 2.4. Experimental Design

First, 50 pfu of MCMV or culture medium as control was injected into caspase-12^−/−^ and control caspase-12^+/+^ mice within 3 days after birth via the intraperitoneal (i.p.) route. At 14 days, 8 months or 12 months post infection (p.i.), the mice were anesthetized, and spectral-domain (SD) optical coherence tomography (SD-OCT), an advanced, non-invasive imaging technique that uses the spectrum of light to quickly create high-resolution, cross-sectional images of the whole retina, was performed and total retinal thickness was measured using the Bioptigen Spectral-Domain Ophthalmic Imaging System (En-visu R2200; Bioptigen, Morrisville, NC, USA) as described previously by our laboratory [41,43]. The mice were euthanized, and the eyes and extraocular tissues including salivary glands, lungs, livers and kidneys were collected and prepared for H&E staining, Western blot, real-time PCR and real-time RT-PCR (qRT-PCR) as described below. A total of 80 mice (20 mice for each group) were used in all experiments.

### 2.5. Western Blot

Western blotting on proteins extracted from MCMV-infected or uninfected caspase-12^−/−^ or control caspase-12^+/+^ eyes was performed as previously described [39,44]. Equal amounts of protein were separated by SDS-PAGE followed by transfer onto polyvinylidene difluoride (PVDF) membranes (GE Healthcare, Piscataway, NJ, USA). Blots were blocked with 5% nonfat dry milk for 1 h at room temperature, then incubated overnight at 4 °C with primary antibody. Following washing, membranes were incubated with HRP-conjugated secondary antibody for 1 h at room temperature. Immune complexes were detected by chemiluminescence (ECL; GE Healthcare) using the ChemiDoc XRS+ blot imaging system (Bio-rad, Hercules, CA, USA). To verify equal loading among lanes, membranes were also probed with anti-β-actin. Band signal intensity was quantified using ImageJ (version 1.54g, National Institutes of Health, Bethesda, MD, USA).

### 2.6. PCR, Real-Time PCR and RT-PCR

All primer sequences used for murine cytomegalovirus genes and rd8 were described previously by our laboratory [41]. DNA was extracted from eyes and extraocular tissues by overnight digestion with proteinase K at 56 °C with continuous vigorous mixing, followed by centrifugation. The supernatant was removed and DNA was precipitated with isopropanol as described before [41]. Total RNA was extracted from eyes using the RNeasy mini Kit (Qiagen, Hilden, Germany) according to the manufacturer’s instructions. Reverse transcription of mRNA was performed on 1 µg of total RNA using the genomic DNA Clear cDNA Synthesis Kit (Bio-Rad, Hercules, CA, USA) according to the manufacturer’s instructions. For real-time PCR or RT-PCR, DNA or cDNA were amplified in a 20 μL reaction consisting of 10 μL 2×SYBR Mix (Bio-Rad), 0.2 μL of 20 pmol/μL primer mixture and 1 μL of DNA (50 ng) or cDNA prepared by reverse transcription (RT) of mRNA on 1 μg of total RNA. Reaction conditions were 5 min at 95 °C followed by 40 cycles of 15 s at 95 °C, 20 s at 60 °C and 20 s at 72 °C. The amount of DNA was represented directly by the Cycle Threshold (Ct) value. mRNA expression levels were represented by the ΔCT and were normalized to the housekeeping gene beta actin.

The expression of 84 key mouse genes involved in antiviral response was analyzed by real-time PCR (RT-PCR; ABI 7900HT, 140 Applied Biosystems, Carlsbad, CA, USA) using the RT2 Profiler™ gene PCR array kit from QIAGEN Inc (Germantown, MD, USA) according to the manufacturer’s instructions. Gene expression data were analyzed using the GeneGlobe (Qiagen)-supported RT^2^ Profiler PCR Data Analysis module online (https://dataanalysis2.qiagen.com/pcr, accessed on 11 November 2021) and gene expression was normalized to the mean of 4 housekeeping genes including Actb, B2m, Gapdh and Hsp90ab1.

### 2.7. Statistical Analysis

All data were expressed as mean values ± SEM (standard error of mean), where *n* is the number of mice or eyes used in each experimental group. Statistical significance was determined using a 2-tailed *t*-test performed by the GraphPad Prism 8 analysis tool. *p* values < 0.05 were deemed significant.

## 3. Results

### 3.1. MCMV Latency in Eyes of Caspase-12^−/−^ and Caspase-12^+/+^ Mice Following Newborn Peritoneal Infection

Our previous studies have shown that during systemic neonatal MCMV infection of BALB/c mice, MCMV disseminates to and remains latent in the choroid and RPE of the eye as well as several systemic organs/tissues including the liver and salivary glands [41]. To determine if depletion of caspase-12 affects MCMV spread and latency, MCMV (50 pfu per mouse) was injected intraperitoneally (i.p.) into caspase-12^−/−^ or caspase-12^+/+^ mice (both on a BALB/c background) at <3 days after birth. This experiment showed that no replicating virus was present in eyes, salivary glands or lungs at both 8 and 12 months p.i., although virus DNA was detected in eyes and extraocular tissues of caspase-12^−/−^ and control mice at the same time points. Although caspase-12 has been shown to play a role in innate immunity against ocular acute MCMV infection, depletion of caspase-12 did not prevent or slow the establishment of MCMV latency in eyes of infected mice. As shown in Figure 1A, similar amounts of virus DNA were detected in tissues from caspase-12^−/−^ and caspase-12^+/+^ mice. As previously observed in our laboratory [41], cytomegalovirus latent ocular infection resulted in the expression of a number of virus genes including early genes (*m18*) and other virus genes that function in either an anti-cell death (m36, m37, m38.5, m41, m45), immune modulation (m04) or protein assembly (m80) capacity in wild-type infected caspase-12^+/+^ mice (Table 1 and Figure 1B). Interestingly, similar levels (Figure 1B) and similar frequencies (Table 1) of MCMV gene expression were detected in latently infected caspase-12^−/−^ eyes compared to latently infected wild-type eyes.

### 3.2. Caspase-12 Depletion Results in a Less Severe Retinal Degeneration in Aged MCMV-Infected Mice

Using a Leica Envisu R2210 system, SD-OCT examinations were performed in latently infected caspase-12^−/−^ and wild-type mice before collection of eyes and extraocular tissues. Retinal thickness was also calculated. Compared to age-matched MCMV-infected wild-type eyes mean retinal thickness was significantly higher in MCMV latently infected, aged caspase-12^−/−^ mice (Figure 2A,B). This observation was also confirmed by H&E staining (Figure 2C).

### 3.3. Analysis of Cell Death Pathways by Western Blot

Our previous studies have shown that caspase-12 contributes to retinal bystander cell death during MCMV retinitis via multiple cell death pathways [40]. Since a less severe retinal degeneration was observed in aged infected caspase-12^−/−^ mice compared to age-matched infected wild-type controls, we hypothesized that caspase-12 might play a role in retinal degeneration resulting from MCMV ocular latency via similar pathways to those involved in retinal bystander cell death during MCMV retinitis. To test this hypothesis, eyes were collected from infected and uninfected caspase-12^−/−^ and caspase-12^+/+^ mice and prepared for Western blots. As shown in Figure 3, key molecules participating in apoptosis (caspase-3 and caspase-8) and necroptosis (RIP3 and MLKL) were activated in infected mice compared to age-matched, uninfected controls, indicating that both apoptosis and necroptosis participate in retinal degeneration during MCMV latency. Since the protein levels of the majority of these molecules were significantly lower in aged infected caspase-12^−/−^ mice compared to age-matched infected wild-type controls, caspase-12 may contribute to retinal degeneration by apoptosis and necroptosis.

### 3.4. p53 and NF-κB Activation

Previous experiments in our laboratory have shown that several pathways involved in innate immunity and inflammation, including the NF-κB [39,45], inflammasome [39,45,46] and p53 [40] pathways, were activated during acute MCMV ocular infection. To determine if these pathways are activated during MCMV latency and if these pathways are affected by lack of caspase-12, the protein levels of NF-κB, p53 and caspase-1 were analyzed by Western blotting. Although caspase-1 was not activated during MCMV latency, both NF-κB and p53 were activated as the protein levels of NF-κB p65 and NF-κB p52 as well as p53 were significantly increased in latently infected eyes, compared to age-matched uninfected control eyes (Figure 4). In addition, caspase-12 contributes to the activation of p53 and NF-κB since the protein levels of NF-κB p65 and NF-κB p52 as well as p53 were significantly reduced in aged infected caspase-12^−/−^ mice compared to age-matched infected wild-type controls (Figure 4).

### 3.5. Analysis of Expression of Inflammatory Genes by Real-Time RT-PCR

Since our previous studies have indicated that MCMV ocular latency is associated with increased in situ inflammation and caspase-12 contributes to p53 and NF-κB activation during MCMV latency, as one of the inflammatory caspases [28,29,30,31,32,33], caspase-12 might play a role in ocular inflammatory responses [41]. To test this possibility, the expression of 84 genes related to immune responses and inflammation was analyzed by real-time RT-PCR in MCMV-infected eyes of both caspase-12^−/−^ and caspase-12^+/+^ mice. As shown in Table 2, among 84 inflammatory genes analyzed, the expression of 32 genes was reduced, whereas only 8 genes showed increased expression in MCMV-infected caspase-12^−/−^ eyes, compared to MCMV-infected caspase-12^+/+^ eyes.

## 4. Discussion

Caspase-12, which belongs to the family of inflammatory caspases, resides at the endoplasmic reticulum (ER), where it is cleaved and activated during ER stress-induced cell death [34,35,36], but is unaffected by other apoptotic stimuli [34]. The human genome encodes either or both the full-length and truncated isoforms of caspase-12, whereas the mouse genome encodes only full-length caspase-12 [47,48]. It is expressed in various organs including the eyes and brain [49,50], and many studies have shown that caspase-12 participates in neural cell death [34,37,51,52,53,54,55,56,57,58,59] and contributes to many neuronal diseases including neurodegenerative diseases [57,58] and brain ischemic injury [37,59], as well as retinal diseases [53,54,55]. A study by Nakagawa et al. indicated that caspase-12 was involved in ER stress-induced apoptotic signals and contributed to Amyloid-beta (Aβ) neurotoxicity in cortical neurons, although it was not essential for apoptosis induced by various types of non-ER stress, such as anti-Fas, staurosporine, or TNF plus cycloheximide in non-neuronal tissues and cells [34]. In addition to ER stress-induced neuronal apoptosis via activation of caspase-3 [56], several non-caspase-mediated mechanisms have also been reported [50,60,61], including one report which indicated that calpain, but not caspases, mediated caspase-12 processing since calpain inhibition rescued cultured microglia from injury induced by oxygen and glucose deprivation (OGD) [50]. Other studies have shown that caspase-12-related cell death was triggered by excessive calcium influx, followed by poly (ADP-ribose) polymerase-1 (PARP1) over-activation, and release of AIF [60,61].

Caspase-12 could play various roles in innate immune responses and inflammation, depending on the type of pathogen and tissue/cell type [28,29,30,31,32,33]. For instance, caspase-12 was shown to inhibit mucosal immunity to bacterial infection via suppression of the Nod-Rip2-NF-kB signaling pathway [33], as well as via inhibition of caspase-1 activation and subsequent IL-1β and IL-18 secretion [28], although the latter remains controversial [29]. Caspase-12 was shown to be activated during respiratory syncytial virus (RSV) and bovine viral diarrhea virus (BVDV) infections [30,31], while activated caspase-12 has been reported to play an important role in controlling West Nile virus (WNV) infection via the pattern recognition receptor RIG-I and subsequent production of type I interferon [32]. In addition, it was reported that in cultured mouse astrocytes ER stress leads to caspase-12 cleavage, upregulation of NLRP3 and NF-kB, caspase-1 activation and IL-1b processing [62]. These studies suggest that ER stress can lead to inflammation via caspase-12 activation [37].

Our previous studies have shown that low levels of caspase-12 are present in the RPE/choroid in normal ocular tissue [40]. During MCMV retinitis, caspase-12 expression is strongly enhanced and spreads to the retina, where it plays a significant role in bystander cell death during MCMV retinitis, since depletion of caspase-12 results in lower levels of cleaved caspase-3 in both the retina and RPE/choroid of MCMV-infected eyes [40]. Moreover, caspase-12 enhanced the innate immune response against acute ocular virus infection in the neural retina but not in non-neuronal ocular tissue including the RPE/choroid, while depletion of caspase-12 was associated with increased virus spread and more severe retinitis [40].

Studies in our laboratory have shown that during systemic neonatal MCMV infection of BALB/c mice, MCMV disseminates to and remains latent in the choroid and RPE. Latent ocular infection combined with ocular MCMV gene expression induces in situ inflammation and development of retinal degeneration in aged mice [41]. The results presented here demonstrate that caspase-12 participates in retinal degeneration related to long-term ocular MCMV latency since mean retinal thickness was significantly greater in MCMV latently infected aged caspase-12^−/−^ mice compared to eyes of age-matched infected caspase-12^+/+^ mice. In addition to caspase-mediated apoptosis, our results indicate that caspase-12 probably also participates in cell death mediated by necroptosis since depletion of caspase-12 resulted in significant reductions in levels of RIP3, p-RIP3 and MLKL. Caspase-12 also contributes to inflammation/innate immunity induced by MCMV ocular latency via the NF-κB pathway since depletion of caspase-12 significantly reduced protein levels of NF-κB p65 and p52 as well as expression of many inflammatory genes in aged latently infected mice. Although depletion of caspase-12 is associated with increased virus spread and a more severe retinitis in immunosuppressed mice following intraocular inoculation of MCMV during acute ocular infection [40], the status of ocular MCMV latency was not affected by caspase-12 since similar levels of MCMV DNA and MCMV expressed genes were detected in latently infected caspase-12^−/−^ eyes compared to latently infected wild-type eyes.

Compared to age-matched uninfected eyes, protein levels of p53 were significantly elevated in latently infected eyes, while depletion of caspase-12 substantially reduced the elevation in p53 levels. Interestingly, protein levels of p53 are also significantly elevated in infected eyes during acute ocular MCMV infection, while depletion of caspase-12 substantially reduced the elevation in p53 levels [40]. As a DNA-binding transcription factor, p53 orchestrates gene expression programs in response to various stresses and regulates several pathways which determine the cell’s fate, whether it be cell cycle arrest or cell death by apoptosis [63]. p53 is not only a major tumor suppressor protein in vertebrates but is also involved in chronic inflammation [64], and plays a significant role in the innate immune response against virus infection [65,66,67] via several pathways including Toll-like receptor 3 (TLR3) [68], IFN regulatory factor 9 [69] and ubiquitin-like protein interferon-stimulated gene 15 (ISG15) [70].

In summary, our results presented herein demonstrate that caspase-12 contributes to retinal degeneration in aged BALB/c mice following systemic neonatal MCMV infection. The mechanism is most likely related to the activation of cell death pathways resulting in apoptosis and necroptosis and increased NFkB activation and subsequent inflammation. However, there are several limitations to this study. Although depletion of caspase-12 is associated with significantly reduced protein levels of key molecules participating in apoptosis and necroptosis, we currently lack direct evidence demonstrating caspase-12 activation of these pathways. In addition, p53 activation and its resulting pathologies vary enormously among tissues [63]. Therefore, further studies are needed to determine how caspase-12 enhances the production of p53 and the exact role played by p53 in inflammation and retinal degeneration during MCMV latency as well as during MCMV acute infection.

## Figures and Tables

**Figure 1 viruses-17-01206-f001:**
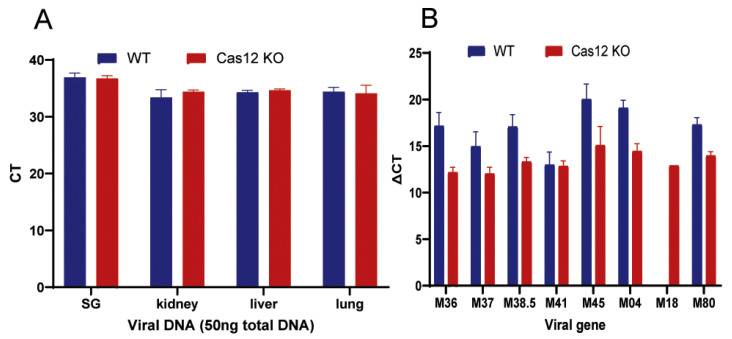
MCMV latency following newborn peritoneal infection. (**A**) Relative levels of MCMV DNA by real-time PCR in salivary glands (SG), kidneys, livers and lungs from caspase-12^−/−^ (Casp12KO) and caspase-12^+/+^ (WT) mice at 8 or 12 months p.i. (**B**) Relative expression levels of mRNA for several MCMV genes by real-time RT-PCR in eyes from caspase-12^−/−^ and caspase-12^+/+^ mice at 8 or 12 months p.i. Data are shown as mean ± SEM (*n* = 6).

**Figure 2 viruses-17-01206-f002:**
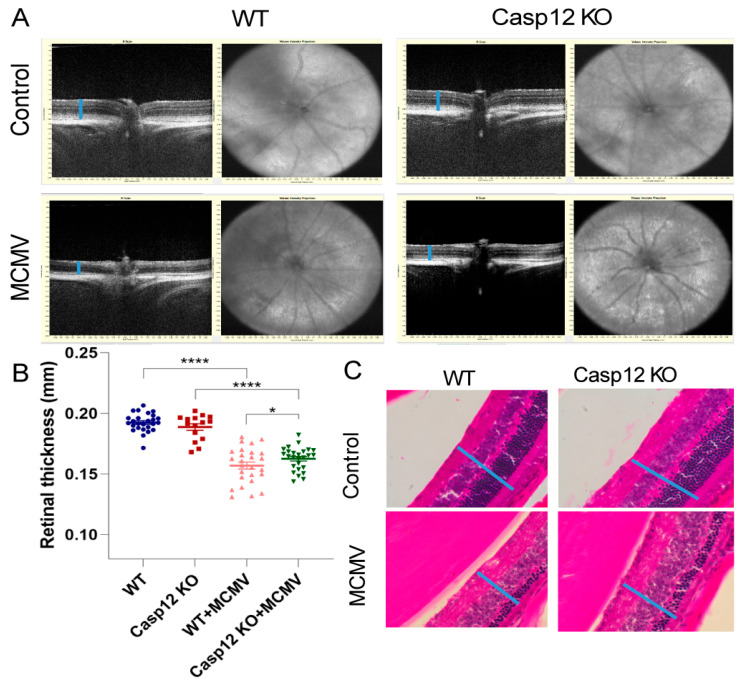
Caspase-12 deficiency alleviates retinal degeneration in MCMV-infected mice at 12 months p.i. (**A**) Representative images of SD-OCT in eyes of latently infected caspase-12^−/−^ and caspase-12^+/+^ mice, together with age-matched control eyes. The retinal layer is indicated by a vertical line. (**B**) Retinal thickness by SD-OCT. * *p* < 0.05; **** *p* < 0.0001 using the Mann–Whitney test (caspase-12 KO group, *n* = 15 mice; other 3 groups, *n* = 26 mice). (**C**) Representative photomicrographs of H&E staining in eyes of latently infected caspase-12^−/−^ and caspase-12^+/+^ mice, together with age-matched control eyes (*n* = 3 mice). The retinal layer is indicated by a vertical line (magnification: 20×).

**Figure 3 viruses-17-01206-f003:**
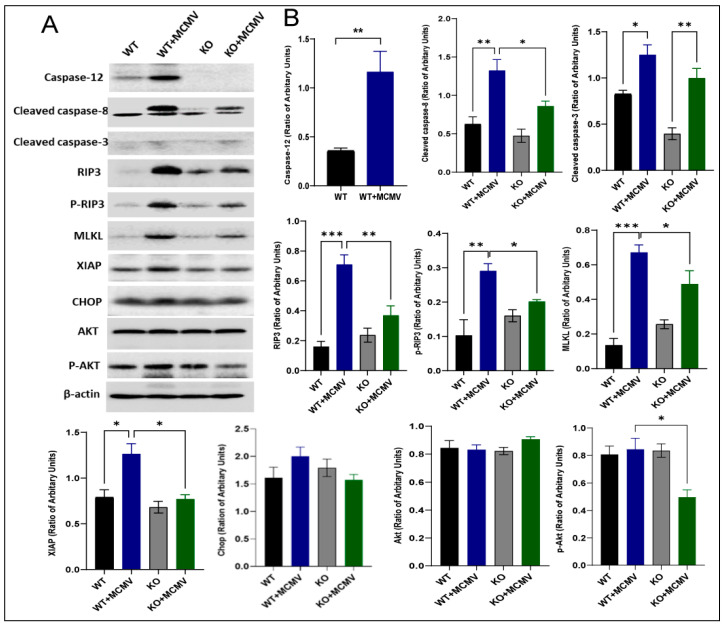
Cell death pathways. (**A**) Western blot of caspase-12, cleaved caspase-8, cleaved caspase-3, RIP3, p-RIP3, MLKL, XIAP, CHOP, AKT and p-AKT, in eyes of latently infected caspase-12^−/−^ and caspase-12^+/+^ mice at 12 months p.i., together with age-matched control caspase-12^−/−^ and caspase-12^+/+^ mice. (**B**) Ratio of caspase-12, cleaved caspase-8, cleaved caspase-3, RIP3, p-RIP3, MLKL, XIAP, CHOP, AKT and p-AKT to β-actin. Data are shown as mean ± SEM (*n* = 3 eyes from 3 mice). * *p* < 0.05, ** *p* < 0.01, *** *p* < 0.001 by 2-tailed *t*-test.

**Figure 4 viruses-17-01206-f004:**
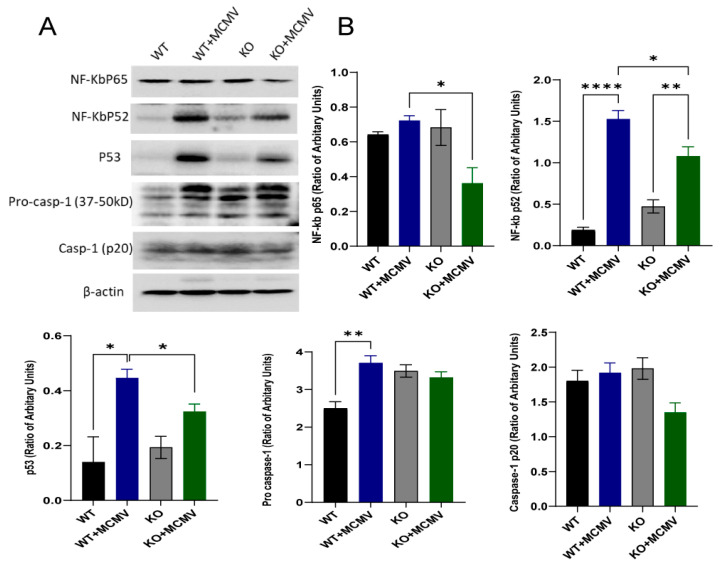
NF-κB and p53 expression. (**A**) Western blot of NF-κB p65, NF-κBp52, p53 and caspase-1 (Casp1) in eyes of latently infected caspase-12^−/−^ and caspase-12^+/+^ mice at 12 months p.i., together with age-matched control caspase-12^−/−^ and caspase-12^+/+^ mice. (**B**) Ratio of NF-κB p65, NF-κBp52, p53, Pro-casp1 and Casp1 (p20) to β-actin. Data are shown as mean ± SEM (*n* = 3 eyes from 3 mice). * *p* < 0.05, ** *p* < 0.01, **** *p* < 0.0001 by 2-tailed *t*-test.

**Table 1 viruses-17-01206-t001:** Frequency of ocular MCMV gene expression measured by real-time RT-PCR.

Gene	Functions	WT	KO
M36	Anti-cell death	5/6	5/6
M37	Anti-cell death	5/6	5/6
M38.5	Anti-cell death	4/6	4/6
M41	Anti-cell death	4/6	3/6
M45	Anti-cell death	2/6	3/6
M04	Immune modulation	2/6	3/6
M18	Early	0/6	1/6
M80	Protein assembly	4/6	2/6

Expression of 8 MCMV genes including early genes (m18) and other virus genes that function in anti-cell death capacities (m36, m37, m38.5, m41, m45), immune modulation (m04) or protein assembly (m80) [41] was measured in eyes of latently infected mice (*n* = 6 mice). Similar frequencies of MCMV gene expression were detected in latently infected caspase-12^−/−^ eyes compared to latently infected wild-type eyes.

**Table 2 viruses-17-01206-t002:** Expression of inflammatory genes in MCMV-infected caspase-12^−/−^ eyes vs. MCMV-infected caspase-12^+/+^ eyes (*n* = 4 eyes).

Increased Expression		Decreased Expression	
Gene Symbol	Fold Regulation	Gene Symbol	Fold Regulation
Atg12	2.47	Atg5	−2.07
Casp1	3.53	Azi2	−4.25
Chuk	3.03	Ccl5	−2.87
Cxcl9	7.35	Ctsb	−4.4
Fadd	2.24	Cxcl10	−2.49
Ifna2	3	Cxcl11	−2.05
Irf5	4.51	Cyld	−2.83
Nod2	4.81	Ddx3x	−3.31
Spp1	3.65	Ddx58	−2.06
Tlr7	5.91	Fos	−3.11
Tlr8	3.66	Hsp90aa1	−3.99
Tradd	2.34	Irak1	−2.61
		Irf7	−4.4
		Isg15	−3.77
		Jun	−2.16
		Map2k1	−3.6
		Map2k3	−10.97
		Map3k7	−3.87
		Mapk14	−2.01
		Mapk3	−3.07
		Mx1	−4.99
		Nfkb1	−8.96
		Nfkbia	−2.68
		Pstpip1	−2.27
		Ripk1	−4.87
		Stat1	−3.27
		Tlr3	−3.02
		Traf6	−2.01
		Actb	−9.03
		Gapdh	−3.36
		Gusb	−2.33
		Hsp90ab1	−3.45

The expression of key mouse genes involved in the antiviral response was analyzed by real-time PCR using the RT2 Profiler™ gene PCR array kit from QIAGEN. The expression of 32 genes was reduced, whereas 8 genes showed increased expression in MCMV-infected caspase-12^−/−^ eyes, compared to MCMV-infected caspase-12^+/+^ eyes.

## Data Availability

The data presented in this study are included in this published article. All the data can be shared upon request by email.

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
