# Peer review of "Depletion of Caspase-12 Alleviates Retinal Degeneration in Aged BALB/c Mice Following Systemic Neonatal Infection by Murine Cytomegalovirus (MCMV)"

_viruses, 2025, doi:10.3390/v17091206_

Round 1

Reviewer 1 Report

Comments and Suggestions for Authors

In their previous publication, the authors reported that BALB/c mice with caspase-12 knockout exhibited more severe retinal pathology 14 days after infection with low-dose CMV (Ref. 40). In the present study, the authors examined infected mice at 12 months of age and analyzed multiple genes and proteins associated with retinal inflammation. The findings further support the role of caspase-12 in ocular CMV infection. This is an interesting study, and several revisions could strengthen the manuscript:

Figure 1 – The method of quantitation is not clearly defined. What does “ΔCt” represent?

Figure 2 / Line 180 – There is an apparent inconsistency: Line 180 states that caspase-12 KO results in less severe retinal degeneration, whereas the Figure 2 title states that caspase-12 KO exacerbates retinal degeneration.

Additional discussion relating to Ref. 40 would help provide context and clarify the findings.

Did the authors examine caspase-12 immunostaining in the retina, either with or without infection?

Author Response

We thank the reviewer for his/her very thorough and insightful comments.  We have listed the reviewers’ comments and questions below with our responses located  immediately following each comment and/or question. Changes within the body of the manuscript are indicated by red font.

Reviewer 1.

In their previous publication, the authors reported that BALB/c mice with caspase-12 knockout exhibited more severe retinal pathology 14 days after infection with low-dose CMV (Ref. 40). In the present study, the authors examined infected mice at 12 months of age and analyzed multiple genes and proteins associated with retinal inflammation. The findings further support the role of caspase-12 in ocular CMV infection. This is an interesting study, and several revisions could strengthen the manuscript:

Figure 1 – The method of quantitation is not clearly defined. What does “ΔCt” represent?

Response: We have added the method of quantitation used in PCR, Real-Time PCR and RT-PCR in section 2.6

Figure 2 / Line 180 – There is an apparent inconsistency: Line 180 states that caspase-12 KO results in less severe retinal degeneration, whereas the Figure 2 title states that caspase-12 KO exacerbates retinal degeneration.

Response: We apologize for this error. The title of Figure 2  has been changed to:  Caspase-12 deficiency alleviates retinal degeneration in MCMV-infected mice at 12 months p.i.

Additional discussion relating to Ref. 40 would help provide context and clarify the findings.

Response: The authors thank the reviewer for the suggestion. Additional discussion relating to ref 40 has been added to the last paragraph of the discussion

Did the authors examine caspase-12 immunostaining in the retina, either with or without infection?

Response: We have tried several antibodies against caspase 12 but none proved reliable for immunostaining.

Reviewer 2 Report

Comments and Suggestions for Authors

HCMV and MCMV encode different homologues of host immune-related genes. These affect disease pathogenesis. Hence the argument that MCMV has been “used extensively” to model the pathogenesis of HCMV disease now needs to be more critically evaluated

Section 2.3 says that caspase deficient “experimental mice” were derived by crossing BALB/c mice with C57BL Caspase-12-/- mice through at least 7 generations. Given the extensive differences between MCMV disease in BALB/c and C57BL mice (organ tropisms, viral persistence, NK responses etc etc), we need some assurance that no C57BL genes remained in the “experimental mice”. 7 generations is minimal for this purpose. Note line 165 claims both had a “BALB/c background” but I don't see evidence of this.

Re Fig 1. I am not sure what units are used but it is odd to see as much MCMV in the kidneys as the SG. Were the adrenal glands and adipose tissues removed? What is meant by “8 or 12 months pi”?

Re Table 1: what organ? What time point? Are these genes homologues of mouse genes?...if so, which mouse genes?

Re fig 2: I cant interpret the images from the information provided. Lines or arrows on the images would help. Why do the MCMV samples in Fig 2C have a pink triangle top left? Also how are 2A and 2C based on n=15 and n=3? Importantly, Fig 2B shows more variation within the groups of infected mice than between these groups. Are the timepoints constant?

Table 2 presents fold regulation as a negative number when expression decreases. Is this usual…I expected to see ratios <1. Transcripts decreased include Gapdh which is often used as a std. So do we believe these findings…it would be good to see confirmatory RT-PCR data for the proteins identified on westerns.

The gist of the data is that a lack of caspase-12 leads to less MCMV replication evident by PCR (Fig 1)…where this aligns with less inflammation (Figs 2 & 3) or at least altered inflammation (Table 2).
Hence either caspase inhibits the virus directly and the reduced/altered inflammation is a consequence

OR

Caspase reduces/alters inflammation as a primary effect and this (perhaps paradoxically) reduces viral replication.

The Discussion tiptoes around this but does not address it

Finally…MCMV appears to reach the eyes but may other organs exhibit persistent pathology (eg: the heart)…are the present results replicated elsewhere?

Author Response

We thank the reviewer for his/her very thorough and insightful comments.  We have listed the reviewers’ comments and questions below with our responses located  immediately following each comment and/or question. Changes within the body of the manuscript are indicated by red font.

Reviewer 2

HCMV and MCMV encode different homologues of host immune-related genes. These affect disease pathogenesis. Hence the argument that MCMV has been “used extensively” to model the pathogenesis of HCMV disease now needs to be more critically evaluated

Section 2.3 says that caspase deficient “experimental mice” were derived by crossing BALB/c mice with C57BL Caspase-12-/- mice through at least 7 generations. Given the extensive differences between MCMV disease in BALB/c and C57BL mice (organ tropisms, viral persistence, NK responses etc etc), we need some assurance that no C57BL genes remained in the “experimental mice”. 7 generations is minimal for this purpose. Note line 165 claims both had a “BALB/c background” but I don't see evidence of this.

Response: we did not check if C57BL genes remained in our “experimental mice”, but according to the colony breeding theory, following a 7 generation back cross, the purity of the Balb/c background should be 99.2188%. Therefore, we consider these mice to be Balb/c mice. Since our control (“wild type”) mice also came from the same back cross, the purity of the Balb/c background should also be 99.2188%, meaning it is identical between the two groups, even  if C57BL genes remained in the experimental mice.

Re Fig 1. I am not sure what units are used but it is odd to see as much MCMV in the kidneys as the SG. Were the adrenal glands and adipose tissues removed? What is meant by “8 or 12 months pi”?

Response: Although virus titer is significantly higher in salivary gland compared to other organs during acute infection, no replicating virus was recovered in any organs including the salivary gland. As shown in Fig 1, similar low level amounts of MCMV DNA were detected in salivary glands and other organs, including kidney, by real time PCR. All adrenal glands and adipose tissues were removed from the kidney. MCMV latency in the kidney has been reported in other publications (Kidney International, 1985;28:922 . Am J Transplant. 2012;12:1024).

Re Table 1: what organ? What time point? Are these genes homologues of mouse genes?...if so, which mouse genes?

Response: The experiments were performed in eye tissues at 12 months post infection. All genes detected are virus (MCMV) genes. To eliminate any possible ambiguity, the title of Table 1 has been changed to “Ocular MCMV gene expression measured by real-time RT-PCR” 

Re fig 2: I cant interpret the images from the information provided. Lines or arrows on the images would help. Why do the MCMV samples in Fig 2C have a pink triangle top left? Also how are 2A and 2C based on n=15 and n=3? Importantly, Fig 2B shows more variation within the groups of infected mice than between these groups. Are the timepoints constant?

Response: The purpose of Fig 2 is to show retinal thickness measured  by OCT and H& E staining. Vertical lines have been added to mark the retinal layer. Fig 2A shows representative results of OCT examinations which were performed in 15 live animals (30 eyes) of each group (n=15).  Fig 2B shows all measurement data resulting from OCT examinations. As shown by the vertical lines in Fig 2A, retinal thickness was increased in infected KO eyes (lower right), compared to infected WT eyes (lower left).  Slides from 3 eyes in each group were stained with H & E (n=3)

Table 2 presents fold regulation as a negative number when expression decreases. Is this usual…I expected to see ratios <1. Transcripts decreased include Gapdh which is often used as a std. So do we believe these findings…it would be good to see confirmatory RT-PCR data for the proteins identified on westerns.

Response: Fold change is a simple ratio of gene expression levels between two conditions, while fold regulation is a more nuanced term that indicates the direction and magnitude of change, often taking into account whether the change is an increase (positive number) or decrease (negative number).

The gist of the data is that a lack of caspase-12 leads to less MCMV replication evident by PCR (Fig 1) where this aligns with less inflammation (Figs 2 & 3) or at least altered inflammation (Table 2). Hence either caspase inhibits the virus directly and the reduced/altered inflammation is a consequence OR Caspase reduces/alters inflammation as a primary effect and this (perhaps paradoxically) reduces viral replication. The Discussion tiptoes around this but does not address it

Response: All experiments were performed in latently infected mice, in which MCMV replication could not be detected. As indicated in the discussion (line 316-319), the status of ocular MCMV latency was not affected by caspase-12 since similar levels of MCMV DNA and MCMV gene expression were detected in latently infected caspase-12-/-  eyes compared to latently infected wild type eyes.  Caspase 12 induces retinal degeneration via activation of cell death pathways and increased NFkB activation/inflammation. We have modified the discussion to clarify this point.

Finally…MCMV appears to reach the eyes but may other organs exhibit persistent pathology (eg: the heart)…are the present results replicated elsewhere?

Response: We agree with the reviewer that other organs (heart or brain) may exhibit persistent pathology.  However, the principal focus of our research is to achieve a better understanding of eye disease and in particular, the possible role of viruses, such as the beta herpesviruses, in this process. As a result, the study of other organs is not within our area of expertise and would probably lead to a dilution of resources and energies needed for our ocular studies.

Round 2

Reviewer 1 Report

Comments and Suggestions for Authors

no further comments

Author Response

Thank you for your support

Reviewer 2 Report

Comments and Suggestions for Authors

This paper still has problems

There are two diagrams labeled figure 2

In the first, there is no explanation of the whole retina scans (A), there are more than 15 dots (B) and the H&E stains are not explained adequately and have no line indicating the point of reading (C). A histological image cannot reflect n=3.

The second Fig 2 does not provide n values.

Fig 4 notes n=3…mice or eyes?

Table 1 needs a legend with a reference to the source of functional data. I cant see how this information is used in the manuscript. Does it illuminate latency?

Table 2 has no legend and the title is muddled.

71 references in as excessive list. The manuscript lacks focus and a statement of the limitations

Were CMV antigen positive cells evident in the retinas?

Comments on the Quality of English Language

Sentence structure needs attention to remove ambiguities

Author Response

We thank the reviewer for his/her very thorough and insightful comments.  We have listed the reviewers’ comments and questions below with our responses located  immediately following each comment and/or question. Changes within the body of the manuscript are indicated by red font.

*There are two diagrams labeled figure 2

In the first, there is no explanation of the whole retina scans (A), there are more than 15 dots (B) and the H&E stains are not explained adequately and have no line indicating the point of reading (C). A histological image cannot reflect n=3. The second Fig 2 does not provide n values.

Response: An explanation of the theory underlying SD-OCT has been added to experimental design line 202-204: spectral-domain (SD) optical coherence tomography (SD-OCT), an advanced, non-invasive imaging technique that uses the spectrum of light to quickly create high-resolution, cross-sectional images of whole retina, was performed and total retinal thickness was measured using the Bioptigen Spectral-Domain Ophthalmic Imaging System (En-visu R2200; Bioptigen, Morrisville, NC, USA) as described previously by our laboratory.

 Fig 2B shows all measurement data resulting from SD-OCT examinations. There are 30 dots in each group since we performed SD-OCT in 15 live animals (30 eyes) in each group (n=15). To eliminate any possible ambiguity, we have changed the “n” value to indicate the number of eyes (n=30 eyes) rather than the number of animals (n=15 mice)

 Fig 2C shows representative results of H& E staining from 3 eyes in each group. Vertical lines have been added to mark the retinal layer. As shown by the vertical lines in Fig 2C, retinal thickness was increased in infected KO eyes (lower right), compared to infected WT eyes (lower left). 

*Fig 4 notes n=3…mice or eyes?

Response: The word “eyes” has been added immediately following  n=3

*Table 1 needs a legend with a reference to the source of functional data. I cant see how this information is used in the manuscript. Does it illuminate latency?

Response: The following legend has been added, together with a reference to the source of the functional data,  under Table 1: Expression of 8 MCMV genes including early genes (m18) and other virus genes that function in anti-cell death capacities (m36, m37, m38.5, m41, m45), immune modulation (m04) or protein assembly (m80) [41] was measured in eyes of latently infected mice. Similar levels of MCMV gene expression were detected in latently infected caspase-12-/- eyes compared to latently infected wild type eyes.

MCMV latency is demonstrated by satisfaction of the following conditions, as described in lines 164 and 180-182, 1. no replicating virus was present in eyes, salivary glands or lungs at both 8- and 12-months post infection; 2.   virus DNA was detected in eyes and extraocular tissues of caspase-12−/− and control mice at the same time points.

*Table 2 has no legend and the title is muddled.

Response: The following legend has been added below Table 2: The expression of 84 key mouse genes involved in the antiviral response were analyzed by re-al-time PCR using the RT2 Profiler™ gene PCR array kit from QIAGEN. Expression of 32 genes was reduced, whereas 8 genes showed increased expression in MCMV infected caspase-12−/− eyes, compared to MCMV infected caspase-12+/+ eyes.

71 references in as excessive list. The manuscript lacks focus and a statement of the limitations

Response: One paragraph containing a summary statement and the limitations of the study has been added to the end of discussion:

In summary, our results presented herein demonstrate that caspase 12 contributes to retinal degeneration in aged BALB/c mice following systemic neonatal MCMV infection. The mechanism is most likely related to the activation of cell death pathways resulting in apoptosis and necroptosis and increased NFkB activation and subsequent inflammation. However, there are several limitations to this study. Although depletion of caspase 12 is associated with significantly reduced protein levels of key molecules participating in apoptosis and necroptosis, we currently lack direct evidence demonstrating caspase 12 activation of these pathways. In addition, P53 activity and the resulting pathologies vary enormously amongst tissues [64]. Therefore, further studies are needed to determine how caspase-12 enhances the production of p53 and the exact role played by p53 in inflammation and retinal degeneration during MCMV latency as well as during MCMV acute infection.

*Were CMV antigen positive cells evident in the retinas?

Response: Our previous studies (ref41) demonstrate that MCMV spreads to and becomes latent only in the choroid/RPE layer (but not the inner retina) of BALB/c mice following systemic neonatal infection.  MCMV antigen was not detected in the retina during latency.